# Parthenogenesis and the Evolution of Anisogamy

**DOI:** 10.3390/cells10092467

**Published:** 2021-09-18

**Authors:** George W. A. Constable, Hanna Kokko

**Affiliations:** 1Department of Mathematics, University of York, York YO10 5DD, UK; 2Department of Evolutionary Biology and Environmental Studies, University of Zurich, CH-8057 Zurich, Switzerland

**Keywords:** anisogamy, isogamy, parthenogenesis, facultative sex, adult sex ratio

## Abstract

Recently, it was pointed out that classic models for the evolution of anisogamy do not take into account the possibility of parthenogenetic reproduction, even though sex is facultative in many relevant taxa (e.g., algae) that harbour both anisogamous and isogamous species. Here, we complement this recent analysis with an approach where we assume that the relationship between progeny size and its survival may differ between parthenogenetically and sexually produced progeny, favouring either the former or the latter. We show that previous findings that parthenogenesis can stabilise isogamy relative to the obligate sex case, extend to our scenarios. We additionally investigate two different ways for one mating type to take over the entire population. First, parthenogenesis can lead to biased sex ratios that are sufficiently extreme that one type can displace the other, leading to de facto asexuality for the remaining type that now lacks partners to fuse with. This process involves positive feedback: microgametes, being numerous, lack opportunities for syngamy, and should they proliferate parthenogenetically, the next generation makes this asexual route even more prominent for microgametes. Second, we consider mutations to strict asexuality in producers of micro- or macrogametes, and show that the prospects of asexual invasion depend strongly on the mating type in which the mutation arises. Perhaps most interestingly, we also find scenarios in which parthenogens have an intrinsic survival advantage yet facultatively sexual isogamous populations are robust to the invasion of asexuals, despite us assuming no genetic benefits of recombination. Here, equal contribution from both mating types to zygotes that are sufficiently well provisioned can outweigh the additional costs associated with syngamy.

## 1. Introduction

Explaining the origin of gamete size differences is a much celebrated success story in evolutionary ecology [1,2,3,4,5]. Under a wide range of conditions [1,6,7,8], isogamy (equal gamete sizes) ceases to be stable and evolves into anisogamy (unequal gamete sizes) via disruptive selection that has its origins in what is essentially a quality–quantity tradeoff. Large zygotes are assumed to survive better than small zygotes, but from any given budget, a parent can produce a smaller number of large than small gametes; thus, quality trades off with quantity. In contexts other than gamete size evolution, ecological situations with a quality–quantity tradeoff (or size-number tradeoff) typically lead to an optimal solution that reflects the best compromise between the two conflicting demands on reproductive success [9,10,11]. The coevolution of gamete sizes when there are two (or more) mating types differs from the simplest settings, as there is now a game-theoretic aspect to the problem: Given that sexual reproduction involves syngamy where two gametes fuse, the door is open to the evolution of microgametes that ‘bypass’ the tradeoff by fusing with a macrogamete, which guarantees that the resulting zygote is large even though the microgamete’s own contribution to size remains meagre.

Given suitable parameter values, evolution of microgametes (sperm) can be a successful strategy even though it comes with a clear cost: if microgamete producers (males) and macrogamete producers (females) are approximately equally abundant, and do not differ greatly in their budget for gamete production, then sperm will vastly outnumber fertilisation opportunities, and most of the produced microgametes are a wasted investment [7]. Investing in large numbers is nevertheless stable, given that other microgamete producers do the same; this remains true, and anisogamy can be stable, whether the reason for failure is not finding fertilisable eggs (gamete limitation scenario [7]) or being outcompeted by others’ sperm either in the context of external fertilisation (gamete competition scenario of [7]) or internal fertilisation [8].

All models are simplifications of reality; this is simultaneously a strength and a challenge. Models do not bring much insight if they are so complex that they do not allow seeing the forest for the trees [12,13,14,15,16], and indeed, a strength of the simplest anisogamy models is their ability to show just how few ingredients are needed to establish disruptive selection on gamete size. At the same time, this means that biases can easily creep in: when deciding what essential ingredients to keep in a model and what to leave out, one is not necessarily informed to equal degree by all phenomena in all relevant taxa [17,18,19].

Here, we focus on relaxing one simplification made by early anisogamy models: that sexual reproduction is obligate, i.e., that gametes can only participate in syngamy — or die. Sexual reproduction is often facultative, particularly so once one gets rid of a bias that overemphasizes vertebrate life cycles [19,20]. Over the entire tree of life, life cycles offer numerous variations on a central theme where lineages can persist a long time without engaging in sex [21]. Indeed, rare facultative sex is a key feature of many isogamous species, with important consequences for their evolution [22,23,24]. It therefore appears necessary to complement the ‘canonical’ anisogamy models with ones that take the diverse alternative routes to fitness into account.

Here, we have chosen to examine one such route in detail, and by coincidence, it is near identical with that of [25], a paper that we only became aware of when finalising ours. Both their work and ours consider that gametes that did not participate in syngamy (sex) may survive to become adults (parthenogenesis). They note, like we do below, that this may lead to unequal sex ratios, assuming that a parthenogenetically produced adult is of the same sex (or mating type) as its parent.

Our treatment below differs somewhat from theirs, in that we model potential survival differences between sexually and parthenogenetically produced juveniles by modifying the size–survival relationships directly (in [25], parthenogenetic survival is a fixed fraction of sexually produced zygote survival). We also provide a complementary approach to theirs when tracking variation in adult sex ratios, including considering that a particular mating type may, via parthenogenesis, spread to be so common that sex (with the now outcompeted mating type) becomes impossible. This leads to de facto asexuality due to lack of diversity in mating types. We also consider another possible route to wholly asexual reproduction: the fate of mutants that inherit the gamete sizes from their parents but only reproduce parthenogenetically. Our analysis shows that macrogametes (which in some contexts are equivalent to eggs) are not the only potential route to asexuality, the smaller microgametes may, under certain assumptions and parameters, have a greater potential to turn a population asexual. This advantage exists even if smaller progeny have lower survival, and can be traced back to the numerical advantage of producing small offspring in great numbers. This helps microgametes to (a) often opt for parthenogenetic reproduction if parthenogenesis is a response to not having found a complementary mating type to fuse with, or (b) spread and outcompete others (assuming the survival penalty of being small is not too large) if parthenogenesis is the result of a mutation impacting reproductive mode.

## 2. Model

We take as inspiration for our model the brown algae *Ectocarpus* [26]. *Ectocarpus* are multicellular with a typically diplohaplontic life cycle in which they can exist in a haploid (gametophyte) or diploid (sporophyte) state. Transitions between these states are instigated by the production of haploid gametes that can be one of two genetically determined self-incompatible mating types [27]. Across species in the *Ecotocarpus*, these gametes can be isogamous and mildly anisogamous, with related brown algal lineages displaying strong anisogamy and even oogamy [28,29], making them an ideal study system for investigations of gamete evolution [30].

Upon encountering a compatible mate, gametes can fuse via syngamy to form zygotes, instigating the diploid sporophytic stage of the life cycle. However, if a compatible mate is not encountered, unfertilised gametes of many *Ectocarpus* species can nevertheless ensure reproductive success by forming a parthenosporophyte that can grow vegetatively, instigating the haploid gametophytic stage of the life cycle [31]. Gamete size is recognised as one of the key factors that may determine whether a gametes are capable of undergoing such parthenogenesis (apomixis), with parthenosporophyte production being restricted to large (female) gametes in many anisogamous species, but also possible in smaller (male) gametes in many species in which anisogamy is mild [32,33]. Interestingly, a recent study in giant kelp (which are closely related to *Ectocarpus* [34]) identified a genetically male strain that was capable of parthenogenesis [35] via gametes that were on average half the size of those produced by females. Taken together, this suggests that at least in principle, the propensity of either sex to reproduce parthenogenically may be more labile than commonly assumed. Categorical descriptions of such possibilities for parthenogenic reproduction in isogamous and anisogamous species across the brown algae can be found in [27,33], while a thorough discussion of empirical results relevant for the following theoretical treatment is provided in [25] for both brown and green algae.

### 2.1. Dynamics within Each Generation

Below, we define the model dynamics in specific detail for each subsequent stage; gamete formation, zygote formation, zygote and parthenosporophyte survival, and finally, forming the next adult generation (see Figure 1 for a graphical overview).

#### 2.1.1. Gamete Formation/Gametogenesis

A total of *A* haploid adults enter each generation. We note in Section 2.1.4, however, that our model could equally well be interpreted as consisting of a mix of diploid (sporophyte) and haploid (gametophyte) adults. Each adult carries a self-incompatible mating type allele, with mating type determined at the haploid stage (e.g., the UV sex-determination systems of green and brown algae [27,36]). In the case of two mating types (which we consider here), we will denote these classes *x* and *y*. We will call these ‘sexes’ in the specific context where we measure the adult sex ratio (ASR), which we denote with *R* and define as the proportion of individuals of class *y* (‘males’). Since it would be cumbersome to replace the term ASR with ‘mating type ratio’ whenever a population has (temporarily or permanently) no marked anisogamy, we use ASR throughout this paper for this quantity, even though the two labels ‘male’ and ‘female’, strictly speaking, only apply once anisogamy has evolved.

Each adult is of a fixed mass *M*, which we consider equivalent to its budget for gamete production, since we assume adults to use all their mass for gamete production once reproduction commences. Generations are thus discrete, and reproduction implies that each adult differentiates into a number of gametes of mass $m_{i}<M$ for $m_{i}\in \left\{m_{x},m_{y}\right\}$. The total number of gametes of each class is then given by $N_{i}\in \left\{N_{x},N_{y}\right\}$ such that
(1)$$N_{i}=\frac{A_{i}M}{m_{i}}\;,$$
where $A_{i}$ is the number of adults of type *i* ($A_{i}\in \left\{A_{x},A_{y}\right\}$) and $\sum_{i\in \left\{x,y\right\}}A_{i}=A$. Equation (Equation 1) is a manifestation of the quality–quantity tradeoff, since we assume, below, that survival is mass dependent.

Note that $m_{x}=m_{y}$ implies isogamy. In the case of anisogamy ($m_{x}\neq m_{y}$), we will refer to smaller gametes as *microgametes* and larger gametes as *macrogametes*.

#### 2.1.2. Zygote Formation

The gamete pool is next subjected to fertilisation dynamics, whereby gametes of opposite mating type encounter each other according to mass action dynamics at a rate *a* (independent of gamete size) to form zygotes [37]. For simplicity, we here ignore gamete mortality during fertilisation (see, for instance, Appendix A). The fertilisation kinetics are allowed to run for a period *T*.

We introduce the dimensionless parameter ϕ=aT to capture the joint effects of gamete encounter rate and fertilisation period. Note that for very high encounter rates (or very long fertilisation periods), ϕ→∞. Under this condition, all macrogametes will be fertilized and only microgametes will remain unfertilized in the gamete pool. Strictly speaking, this assumes that the adult sex ratio is not massively female biased, as an overabundance of adult macrogamete producers could, in principle, make microgametes be in short supply; as we will see, however, the relevant scenarios instead are subject to positive feedback where an initial microgamete overabundance leads to a male-biased ASR, yielding even more ’surplus’ microgametes in the next generation; we therefore do not observe female-biased ratios.

#### 2.1.3. Zygote and Unfertilized Gamete Survival

In line with previous models [2,7], we assume that the probability of zygote survival at the end of the fertilisation period is given by a decaying exponential as a function of inverse zygote size ($m_{x}+m_{y}$) with a parameter $\beta_{z}$;
(2)$$\begin{array}{ccc} S_{z}\left(\beta_{z},m_{i},m_{j}\right) & = & e^{-\beta_{z}/\left(m_{i}+m_{j}\right)}\;. \end{array}$$

This is also known as the Vance survival function [38,39]). Selection favours larger zygotes when $\beta_{z}$ is large.

In contrast to previous approaches, but in line with [25], we assume that unfertilised gametes (those that remain in the gamete pool at the end of the fertilisation period) can become a haploid parthenosporophyte and develop into adults [40]. The probability of this succeeding is modelled with an equivalent function as Equation (Equation 2) but now with a parameter $\beta_{p}$;
(3)$$\begin{array}{ccc} S_{p}\left(\beta_{p},m_{i}\right) & = & e^{-\beta_{p}/m_{i}}\;. \end{array}$$

Biologically, we can distinguish between two scenarios. The values of $\beta_{p}$ and $\beta_{z}$ relate to how challenging it is to survive; a large value implies that the juvenile needs to be large for survival to be reasonably high. The ‘parthenogenetic disadvantage’ case has $\beta_{p}>\beta_{z}$, which implies that for any juvenile size, it is more challenging for a parthenogenetically produced juvenile to recruit into the adult population than for a sexually produced zygote to do so. Although [25] model this in a different way (a fixed survival reduction for parthenogens), they consider the parthenogenetic disadvantage scenario the more likely one, and restrict their analysis to this case. However, we additionally analyse the ‘parthenogenetic advantage’ case ($\beta_{p}<\beta_{z}$), where zygote formation is associated with intrinsic costs. Our formulation of this cost implies that zygotes can survive equivalently well as parthenogens, but this requires zygote size to exceed that of parthenogens (note that zygotes combine the mass of two gametes).

At first sight, the parthenogenetic advantage scenario may appear an unlikely one: its assumption of higher parthenogen survival is at odds with parthenogenesis appearing to be conducted as if it was the best of a bad job, i.e., only those who fail to find a partner to fuse with start developing parthenogenetically. We believe, however, there to be several good reasons to include this case in the analysis: (i) zygotes indeed are larger than microgametes (which in turn are the gamete type easily failing to fuse), thus realized survival for zygotes may empirically appear larger [32] without there being an intrinsic penalty for parthenogens per se-our modelling approach allows disentangling size-based advantages and those linked to reproductive mode; (ii) syngamy comes with its own risks [41,42] including, but not limited to, having to express a genome that merges two different parental organisms, and parthenosporophytes may generally experience some growth advantage (e.g., efficient exploitation of low-resource environments [43]); and (iii) one of our model’s goals is to see under what conditions mutations to asexuality can spread, and it appears intuitively clear that such mutations should be beneficial if sex comes with a clear disadvantage —however, this intuition needs to be checked against actual model output.

We also note that our formulation is distinct from other approaches in which gamete survival is modelled as a precursor to fertilisation [2,44], as opposed to a route to parthenogenesis.

#### 2.1.4. Forming the Next Adult Generation

Zygotes and parthenogens survive, according to Equations (Equation 2) and (Equation 3), respectively, to become the new adult generation. Since we assume that adults are haploid, we assume fertilisation is followed by meiosis to yield this outcome (as in e.g., *Chlamydomonas*). The adult population is thus a mixture of individuals that represent unfertilized parthenosporophytes (inheriting the genetic identity of their parent with certainty), and those that are haploid products of meiosis that occurs after fertilisation (inheriting the gamete-size-linked mating type of either parent with a probability of a half; this corresponds to the term 1/2 in Equation (Equation 1) of [25]). Our model can thus be considered a stylized version of the isogamous green algae *Chlamydomonas reinhardtii* [45] or the anisogamous green algae *Volvox carteri* [46], where new adults are categorisable as the mitotic progeny of cells reproducing clonally (having passed through the parthenogenic survival route characterised by Equation (Equation 3) or the meiotic progeny of zygospores (having passed through the zygotic survival route characterised by Equation (Equation 2).

It is important to note that allowing parthenogenesis opens up the possibility of deviations from a 1:1 sex ratio (or, more generally, mating type ratio). Obligately sexual reproduction with Mendelian inheritance that is followed by diploid zygotes splitting into two haploid adults would keep this ratio intact, but this equity is broken when parthenosporophytes from the gamete pool at the end of the fertilisation period, with a likely overabundance of microgametes, contribute to the genetic make up of the next generation.

Our survival functions are not built with a guarantee that exactly *A* adults will survive to form the next generation. We therefore next assume density-dependent regulation, such that the total number of adults equals *A*, without changing the ratios of any category of individuals (implicitly, if the number of haploid adults falls below *A*, we assume vegetative reproduction until *A* exist, while if the population exceeds *A*, the excess is culled without changing the ratios of different mating types present).

Finally, we note that although for simplicity we have assumed an entirely haploid adult population (in line with the reproductive biology of unicellular green algae, such as *C. reinhardtii* [45]), our model could be equally well applied to the diplohaplontic life cycle of *Ectocarpus* if we additionally assume no selective differences between haploid gametophyte and diploid sporophyte adults. This latter scenario merely amounts to a reinterpretation of the products of zygotes, with the frequency of mating type alleles (the key quantity of study) unchanged.

### 2.2. Invasion Dynamics

#### 2.2.1. Evolution of Gamete Sizes

We first investigate the invasion dynamics of mutants that inherit the same self-incompatibility behaviour as their ancestor (i.e., do not change their mating type), but producing gametes of a novel size. We will denote such mutants by $\hat{x}$ or $\hat{y}$. In the following description we will only consider the case of a mutant *y* mating type, $\hat{y}$, for simplicity. Note that equivalent expressions for a population with a mutant *x* mating type, $\hat{x}$, can be obtained be swapping indices for *x* and *y*. This means that our analysis loses no generality: it will be able to track the sizes of both micro- and macrogametes during divergent selection. Note, also, that we will plot all our figures allowing either *y* or *x* to produce larger gametes, thus the convention that *y* produces microgametes is only used in the context of explaining some equations via the most familiar labelling.

Consider a mutant that forms gametes of a size ${\hat{m}}_{y}=m_{y}+\delta m$, where δm may be positive or negative, and $\left|\delta m\right|$ defines a fixed mutational step size. Analogous to the residents, the total number of mutant adults at the start of a generation is denoted by the variable ${\hat{A}}_{y}$ and the total number of mutant gametes by ${\hat{N}}_{y}={\hat{A}}_{y}M/\left(m_{y}+\delta m\right)$ (see Equation (Equation 1)). The survival probabilities of zygotes that result from the fusion of *x* and $\hat{y}$ gametes, and $\hat{y}$ parthenosporophytes, can be be determined from Equation (Equation 2) and (Equation 3), respectively.

Following the introduction of an initial number of mutant adults ${\hat{A}}_{y}=1$, we iterate the dynamics described in Section 2.1 (see Appendix C). We will see that eventually the number of mutant adults either dies out (${\hat{A}}_{y}\to 0$) or grows to displace the ancestral mating type ($A_{y}\to 0$). In the latter case, the gamete sizes present in the population have changed, and the invasion of the mutant *y* mating type can also lead to a new adult sex ratio (ASR). Since we have assumed, for illustration, that $m_{x}>m_{y}$ (i.e., that *y* is the microgametic type), we shall here denote this ratio of microgamete producers (“males”) to microgamete producers (“females”) as *R*, such that
(4)$$R=\frac{A_{y}+{\hat{A}}_{y}}{A_{x}+A_{y}+{\hat{A}}_{y}}\;.$$

The finiteness of the adult population can mean that one mating type (the one that is abundantly produced due to its small size) may, assuming sufficiently high survival to yield numerous adults, outcompete the complementary mating type and thus destroy its own chances of ever engaging in sex. R→1 is indicative of such outcomes, where the population has shifted to de facto asexuality due to loss of mating type diversity.

#### 2.2.2. Mutations to Obligate Asexuality

We will also investigate the possibility of asexuals invading the population. This is a particularly pertinent question given that we consider both disadvantageous and advantageous scenarios for parthenogenesis with respect to survival, as well as microgamete producers being potentially able to outcompete macrogamete producers (which may select for foregoing sex in the first place). Asexual mutants initially possess the same size strategy as their parent, but will not participate in mating dynamics and zygote formation; all asexuals attempt survival via the parthenogenetic route. We will denote such mutants by the index *a*. Asexuals produce progeny of size $m_{a}=m_{x}$ or $m_{a}=m_{y}$ depending on the identity of their ancestor (of mating type *x* or *y*).

Following the introduction of an initial number of ${\hat{A}}_{a}=1$ asexuals, we iterate the dynamics described in Section 2.1 (see Appendix F). Note that because the asexuals experience modified kinetics in which they do not take part in fertilisation (see Appendix B), the only way for asexuals to propagate is through survival of their parthenogenic form. We will see that there are only two possible fates for obligate asexuals: eventually the number of asexual mutants dies out (${\hat{A}}_{a}\to 0$) or grows to displace both of the resident mating types (${\hat{A}}_{a}\to A$).

### 2.3. Evolutionary Dynamics

We assume that mutations occur at a very low rate, such that each invasion has time to complete before another mutation arises. For simplicity, as well as consistency with earlier studies, we assume that mutation occurs at a fixed rate per adult in the population (an alternative, which we do not adopt here, would be to employ a per-gamete mutation rate, which would substantially boost the overall number of mutations brought in by microgametes due to their larger number). Note that while we assume the larger number of gametes produced by microgamete producers (relative to macrogamete producers) does not directly translate into a larger rate of mutations in the former, it is still possible that microgametes are responsible for more mutation events as a whole, in cases where microgamete producers become common in the population due to prolific parthenogenic reproduction. In other words, the sex towards which the ASR is biased is also a larger source of mutations. We pick mutations probabilistically, and follow the subsequent stochastic evolutionary trajectories.

#### 2.3.1. Mutations to a Different Gamete Size

We first ignore mutations to asexuality and focus on the case in which mutations impact gamete sizes. We explain the method by assuming, purely for illustration purposes, that the mutation occurs in an ancestor that is of type *y* (the method is wholly equivalent for *x*). The mutation does not change the mating type; the mutant will still mate with the complementary mating type *x*. The mutant produces either larger (${\hat{m}}_{y}>m_{y}$) or smaller (${\hat{m}}_{y}<m_{y}$) gametes than its ancestor, each scenario occurring with a probability 1/2. The size difference between the mutant and its ancestor is given by |δm| (e.g., ${\hat{m}}_{y}=m_{y}\pm \delta m$). However, we impose a minimum gamete size, $m_{\min}=\left|\delta m\right|$, below which we assume gametes to be inviable, such that mutations to state ${\hat{m}}_{x}<m_{\min}$ or ${\hat{m}}_{y}<m_{\min}$ can be ignored.

The subsequent invasion dynamics are then evaluated according to the description in Section 2.2.1. If the invasion is unsuccessful, the resident population remains unchanged. If the invasion is successful, the resident population is appropriately updated (see Appendix E). In either case, another mutation is then selected randomly to occur, in either *x* or *y* adults proportional to their ratio in the adult population.

We repeat the process described above until the population reaches an evolutionary stable state from which it cannot leave. Note that de facto asexuality may emerge without any asexual mutant having arisen; highly skewed sex ratios can potentially drive one of the mating types to extinction, leaving the remaining mating type unable to find mates, and all reproduction is subsequently parthenogenetic.

#### 2.3.2. Mutations to Asexuality

We next consider the the case in which the mutation is a switch to asexuality, again supposing for illustration that the ancestor is of type *y*. The mutant switches to obligate parthenogenesis which is inherited by all its offspring; gamete size is not impacted by this mutation, but may be subject to further mutations in later generations. The mating type remains unchanged; however, it is no longer expressed in any relevant context (since the mutants and its descendants are obligately asexual). We will assume that mutations to asexuality are less frequent than those affecting gamete size, as this allows us to focus primarily on the invasion potential of asexuals in populations that are an evolutionary stable state with respect to isogamy or anisogamy as described in Section 2.3.1. Further, we do not consider back mutation (from asexual to sexual reproduction) or the evolution of novel mating types or other modifications to self-incompatibility.

The subsequent invasion dynamics are evaluated according to the description in Section 2.2.2. Again, if the invasion is unsuccessful, the resident population remains unchanged. If the invasion is successful, then the resident sexual types are lost, leaving only the asexual parthenogens. Note that even if an asexual lineage has taken over, it does not necessarily (yet) represent a stable situation with respect to the size of its progeny; its sexual ancestry may be visible and further size mutations may be necessary to optimize reproduction in the face of a quality–quantity tradeoff (see Appendix G). Therefore, we do not stop tracking the invasion as soon as asexuals have replaced sexuals, but track the subsequent evolution of the population until it reaches its evolutionary stable state.

## 3. Results

This section is organised as follows. For orientation, we first show examples of evolutionary trajectories (Section 3.1). Section 3.2–Section 3.5.2 seek to quantify these dynamics mathematically, under the assumption that the fertilisation period is very long (ϕ→∞) and that mutations conferring obligate asexuality are absent (see Section 2.3.2). The first of these restrictions means that we restrict our analysis to the regime of gamete competition (under which all macrogametes are fertilized and there is competition for fertilisation among the microgametes, some of which remain unfertilized) and ignore the regime of gamete limitation (under which fertilisation is inefficient and both macrogametes and microgametes do not achieve 100% fertilisation success [47]) [7,25,48]. Meanwhile, we release the second of these restrictions in Section 3.6, in which we investigate the invasion potential mutations to obligate asexuality.

### 3.1. Examples of Evolutionary Strategies

Mutations to obligate asexuality are not necessary for a population to switch to wholly parthenogenetic reproduction. In the absence of such mutations, a long fertilisation period may yield cases where de facto asexuality evolves because microgametes, large enough to survive as parthenogens, take over (Figure 2). In the depicted examples, the ASR is kept ultimately at 1:1 (despite short-term fluctuations) when parthenogenesis brings about a survival disadvantage, while an advantage to parthenogenesis permits de facto asexuality (Figure 2c) but also a variety of other evolutionary stable states, dependent on initial conditions; these include anisogamy (dotted lines), isogamy (dashed lines) and macrogamete extinction (solid lines). In the following sections we seek to quantify this behaviour, and the conditions under which each scenario might occur, mathematically.

### 3.2. Evolutionary Dynamics under Obligately Sexual Reproduction

It is instructive to examine the behaviour of the system when we assume $\beta_{p}\to \infty$, which makes parthenogenesis an unprofitable option (equivalent to assuming *c* = 0 in the notation of [25]). In this case, and also focusing on the case (which we do throughout) that ϕ→∞, the evolutionary dynamics of $m_{x}$ and $m_{y}$ are given by
(5)$$\begin{array}{ccc} \frac{dm_{x}}{d\tau } & = & \frac{\beta_{z}m_{x}-{\left(m_{x}+m_{y}\right)}^{2}}{m_{x}{\left(m_{x}+m_{y}\right)}^{2}} \end{array}$$
(6)$$\begin{array}{ccc} \frac{dm_{y}}{d\tau } & = & \frac{\beta_{z}m_{y}-{\left(m_{x}+m_{y}\right)}^{2}}{m_{y}{\left(m_{x}+m_{y}\right)}^{2}}\;, \end{array}$$
where τ is a rescaled time variable that takes account of the mutation rate (see Appendix E). A phase portrait of this system is given in Figure 3a. We see this displays the expected qualitative dynamics, with unstable fixed points at (0,0) and $\left(\beta_{z}/4,\beta_{z}/4\right)$ (isogamous states) and stable fixed points at $\left(\beta_{z},0\right)$ and $\left(0,\beta_{z}\right)$ (extremely isogamous states, in which one of the gametes is infinitely small), in agreement with earlier models [2,7].

### 3.3. Evolutionary Dynamics of Progeny Size under Asexuality

If one of the two mating types is lost from the population, reproduction in the remaining mating type is restricted to occurring only through parthenogenesis. This can occur if sex ratios become sufficiently skewed (e.g., Figure 2d, solid lines), or if $\beta_{z}\to \infty$ (without zygote survival, selection for gamete sizes becomes frequency independent and only one type survives). The evolutionary dynamics for the gamete size, *m*, of the remaining mating type are then given by
(7)$$\frac{dm}{d\tau }=\frac{\beta_{p}-m}{m^{2}}\;,$$
(see Appendix G). The parthenogens will evolve towards an evolutionarily stable progeny size of $m^{*}=\beta_{p}$ (see red dashed lines, Figure 3b).

### 3.4. Adult Sex Ratio

Above, we showed that fully sexual reproduction maintains an adult sex ratio of 1:1 under our assumptions, while parthenogenesis leads to 100% prevalence of one mating type. Intermediate regimes are more interesting, and we now turn to our attention to them.

We begin by assuming that fertilisation periods are very long, such that ϕ→∞. Note that this means that all macrogametes are fertilized at the end of the fertilisation period, and thus only microgametes (which remain in the gamete pool) are available for parthenogenesis.

We now assume that *y* is the microgametic type ($m_{x}>m_{y}$). The adult sex ratio in the population of residents (i.e., in the absence of mutants, see Equation (Equation 4) can then be expressed as
(8)$$\left.R=\left\{\begin{array}{cc} \frac{m_{y}\left(e^{\frac{\beta_{p}}{m_{y}}}-e^{\frac{\beta_{z}}{m_{x}+m_{y}}}\right)}{2m_{y}e^{\frac{\beta_{p}}{m_{y}}}-\left(m_{x}+m_{y}\right)e^{\frac{\beta_{z}}{m_{x}+m_{y}}}} & \mathrm{if}\;\beta_{p}>\frac{\beta_{z}m_{y}}{m_{x}+m_{y}}-m_{y}\log\left(\frac{m_{y}}{m_{x}}\right)\\ 1, & \mathrm{otherwise} \end{array}\right.\right\}\;\mathrm{for}\;m_{y}<m_{x}$$
(see Appendix D for full derivation).

We now summarize some of the limiting behaviour of Equation (Equation 8). Under isogamy ($m_{x}=m_{y}$), we find unbiased sex ratios (R=1/2). The same result arises in the limit of fully sexual reproduction (e.g., $\beta_{p}\to \infty$), while in the limit of fully parthenogenic reproduction (e.g., $\beta_{z}\to \infty$), we see extinction of the macrogametic sex (R=1). Macrogametic producers can also become extinct at intermediate $\beta_{z}$ for particular values of $m_{x}\neq m_{y}$ (see deeper blue regions in Figure 4b). Note that the macrogamete producers cannot outcompete the microgamete producers while the reverse is possible, which is a direct consequence of our assumption that ϕ→∞ (at the end of the mating dynamics there are no macrogametes left to develop parthenogenetically).

### 3.5. Evolutionary Dynamics: Mixed Sexual and Parthenogenic Reproduction

When both sexual and parthenogenic reproduction are potentially available routes to the next generation, the ASR (see Equation (Equation 8)) plays a crucial role. Not only does it change the rate at which micro- or macrogametes of novel sizes appear in the population (because more mutations occur in individuals of the more common type); it also opens up the possibility of a mating type class (and thus the ability for sexual reproduction) being lost from the population. We will see that these combined effects complicate the straight-forward behaviour predicted in Equation (Equation 5)–(Equation 7).

We again assume that fertilisation periods are very long (ϕ→∞), *y* is the microgametic type ($m_{x}>m_{y}$), and now we additionally focus on the general situation where that both *x* and *y* mating types are present in the population (1>R>0). Under these conditions, we show in Appendix E that the evolutionary dynamics of $m_{x}$ and $m_{y}$ are given by
(9)dmxdτ=βzmx−(mx+my)2myeβp/my−mxeβzmx+mymx(mx+my)22myeβp/my−(mx+my)eβzmx+mydmydτ=1my2(mx+my)22myeβp/my−(mx+my)eβzmx+my×eβzmx+myeβzmx+mymy2eβp/myβzmy+(mx+my)2
(10)+eβzmx+myeβzmx+myeβzmx+my(mx+my)2βpmx+my2−βpmy−βzmxmy2,
where once again τ is a rescaled time variable that takes account of the mutation rate. Note that these equations are derived assuming $m_{x}>m_{y}$ (i.e., that *y* is the microgamete producer). Equivalent expressions for when $m_{y}>m_{x}$ (i.e., when *x* is the microgamete producer) can be obtained by swapping *x* and *y* indices.

The dynamics of Equation (Equation 9) and (10) are summarised in Figure 4, which we can see recapitulate the results of our example simulation trajectories in Figure 2 (see also Figure A3). We see that under parthenogenetic disadvantage ($\beta_{p}>\beta_{z}$), the dynamics show no qualitative differences to the fully sexual case with respect to the fixed points; anisogamy remains a common outcome. There are some quantitative differences near the fixed points: skewed sex ratios generate deviations in evolutionary trajectories (compare Figure 3a and Figure 4a), but still isogamy remains the unstable and anisogamy retains its status as the stable final evolutionary state. These results are also similar to those of [25]. However, when the population is evolving far from these evolutionary states or, alternatively, if we make the alternative assumption of parthenogenetic advantage ($\beta_{p}<\beta_{z}$), we observe more drastic effects of facultative sex (i.e., qualitative departures from the fully sexual case). We will discuss these departures below.

#### 3.5.1. De Facto Asexuality due to Extinction of Macrogametes

In Figure 4b, where parthenogens enjoy a survival advantage ($\beta_{z}>\beta_{p}$), regions of state space emerge in which the sex ratio is so severely skewed that macrogametes are driven extinct (inside the red bounded regions). Once the microgamete producers are the only remaining mating type, they are free to evolve towards the optimal independent cell size identified in Section 3.3 (i.e., $m^{*}=\beta_{p}$, see red dashed lines in Figure 4b). While it is tempting to emphasize the wonderful weirdness of this result by stating that ‘sperm drove eggs extinct’, this mental image misrepresents the dynamics. The microgametes obviously were at no point along the evolutionary trajectories so small that their viability for parthenogenetic reproduction was too severely compromised, otherwise they would not have been able to take over the population of *A* adults. The result is better characterized as mild anisogamy ratios easily leading to a situation where the smaller of the two gamete types evolves to utilize the parthenogenetic route of development.

Note that the prospects of microgametes taking over the entire population depend on a suitable combination of their number (which should be large, so that many remain outside syngamy) and size (which should be large enough for many to survive). While intuition suggests that the product of leftover microgamete number and their viability exceeds the number of surviving zygotes more easily if $\beta_{z}>\beta_{p}$ (i.e., under parthenogenetic advantage), this is not a strict requirement for the eventual extinction of macrogametes. Macrogamete extinction can extend to a region with parthenogenetic disadvantage ($\beta_{p}>\beta_{z}$) so long as $m_{x}$ and $m_{y}$ are sufficiently large (illustrated in Figure A2, which re-plots Figure 4 over a larger region of state space i.e., the $m_{x}-m_{y}$ plane).

An interesting case arises should a population face the need to evolve smaller gamete sizes from a state of isogamy in which $m_{x}=m_{y}\gg \beta_{z}$. Here, should sex still be possible at the endpoint of the evolutionary trajectory, the population must tread a precarious path towards smaller gamete sizes; both mating types should remain, at all times, very close to each other in size, otherwise the adult sex ratio reflects asymmetric gamete numbers in a manner that predicts macrogamete extinction. In other words, while a continuous trajectory exists along which isogamy is maintained (pulling the population towards the point $m_{x}=m_{y}\approx \beta_{z}/4$), any slight deviation from isogamy leads to a skewed adult sex ratio, in which microgametes dominate. This in turn leads to a positive feedback (with further mutations more likely to occur on the microgametic mating type) that drags evolutionary trajectories towards only one mating type surviving, and de facto asexuality.

If the evolutionary trajectory manages to survive this perilous corridor such that macrogamete extinction is avoided (or if instead the evolutionary trajectory starts in an arbitrary anisogamous state), the final state towards which the system evolves varies in a non-trivial way based on the model parameters and initial conditions. We discuss these varied outcomes in the following section.

#### 3.5.2. A Novel Route to Stable Isogamy

In cases of parthenogenetic disadvantage ($\beta_{p}>\beta_{z}$), our model is in line with earlier work [25]: if both mating types are still present in the population, anisogamy is stable. When parthenogenetic reproduction is inherently advantageous (in the sense of $\beta_{z}>\beta_{p}$), the anisogamous states remain stable. However, this change in parameters now adds the isogamous state at $m_{x}=m_{y}\approx \beta_{z}/4$ to the list of alternative stable equilibria. Whether the population evolves towards anisogamy or isogamy depends on initial conditions (see Figure 4b).

If the ancestral population consists of large isogamous gametes ($m_{x}=m_{y}\gg \beta_{z}/4$), then provided the population can survive the perilous corridor of potential macrogamete extinction (see Section 3.5.1), the evolutionary endpoint is typically stable isogamy; the same is true for populations that begin in a state consisting of small isogamous gametes ($m_{x}=m_{y}\ll \beta_{z}/4$). However, stochastic morphological mutations from the isogamous state with small gametes are more likely to push the system into the basin of attraction of the stable anisogamous states.

The basin of attraction for the stable anisogamous states is difficult to define mathematically. However, we can qualitatively argue that it is generally encompassed regions of anisogamy ($m_{x}>m_{y}$ or $m_{y}>m_{x}$) in which the microgamete size is smaller than that found in the stable isogamous state ($\left(\beta_{z}/4\right)⪆m_{y}$ or $\left(\beta_{z}/4\right)⪆m_{x}$, respectively).

### 3.6. Fates of Mutations to Asexuality Depend on the Mating Type They Arise in

In the previous sections we have seen that parthenogenetic advantage ($\beta_{z}>\beta_{p}$) can lead to a variety of non-trivial evolutionary trajectories : the multiple possible end states including 100% ASR following the extinction of macrogametes, stable anisogamy, and stable isogamy. However, as this parameter regime implies some advantage to parthenogenesis, a natural question to ask is whether sexual reproduction itself is here stable against the invasion of asexuals. The section below shows that the particular conditions that allow asexual reproduction to invade depend on whether the mutation to asexuality occurs in an *x* or *y* individual. There are conditions where mutations to asexuality are more successful if they arise in a microgamete producer, or in a macrogamete producer.

We consider microgametes first. For notational simplicity, we assume that the sex that currently produces microgametes is *y* (i.e., $m_{x}>m_{y}$); however, note that the analysis itself does not use this assumption: all figures are symmetric since a particular combination of $m_{x}$ and $m_{y}$ has dynamics that is simply the mirror image of one with $m_{y}$ and $m_{x}$, respectively, taking the same values.

In Appendix F, we show that asexual microgametes (with the mutation arising on a *y* background) can invade and displace their sexual ancestors if
(11)$$\begin{array}{c} \frac{\beta_{z}m_{y}}{m_{x}+m_{y}}-m_{y}\log\left(\frac{m_{y}}{m_{x}}\right)>\beta_{p}\;,\;\mathrm{for}\;m_{y}<m_{x}\;. \end{array}$$

This condition (Figure 5, purple regions) is coincident with the region of state space in which sex ratio distortion (in the absence of mutations to asexuality) leads to microgametes taking over the population (see Equation (Equation 8)). Within this region, de facto asexual and genotypically asexual microgametes behave identically; in the absence of macrogametes, any microgametes (whether or not they aim for sexual reproduction) ultimately attempt survival as parthenogens. In the face of recurrent mutations to asexuality (and no back mutations), genotypic asexuality will eventually reach fixation through drift alone. Although not included in our model, there may also be selection to improve asexual performance (e.g., loss of vestigial traits related to seeking gametes to fuse with).

We next derive the equivalent expression for macrogametes. In this case, there is no equivalent prediction for when sex ratio distortion alone would make macrogamete producers outcompete microgamete producers (macrogametes, being produced in smaller numbers than microgametes, do not experience lack of opposite-sex gametes in our model, and we therefore do not observe positive feedback favouring female-biased sex ratios until microgametes are lost). This does not prevent there being conditions where a mutation that makes a macrogamete producer asexual can spread in a population. In Appendix F, we show that the condition for a macrogametic asexual type to invade is given by
(12)$$\begin{array}{c} \frac{\beta_{z}m_{x}}{m_{x}+m_{y}}>\beta_{p}>\frac{\beta_{z}m_{y}}{m_{x}+m_{y}}-m_{y}\log\left(\frac{m_{y}}{m_{x}}\right)\;,\;\mathrm{for}\;m_{y}<m_{x} \end{array}$$
where the second inequality for $\beta_{p}$ comes from the fact that the ASR must be such that at least some macrogametes are present in the population (i.e., R<1) for an asexual macrogamete mutant to be possible (see Equation (Equation 8)).

We see that asexual macrogametes can displace their sexual ancestors when their size exceeds that of microgametes by a critical amount (orange regions in Figure 5): $\left[\left(\beta_{z}-\beta_{p}\right)/\beta_{p}\right]m_{x}>m_{y}$. Perhaps counterintuitively (should one employ the biological intuition that eggs can be parthenogenetic but sperm hardly so), macrogamete-driven asexuality is absent in Figure 5a where parthenogens are at a disadvantage, while microgamete-driven asexuality is possible whether parthenogens are favoured (Figure 5a) or disfavoured (Figure 5b) over zygotes when surviving to become adults. On the other hand, macrogametes have the property that the parameter region from where they can invade (under parthenogenic advantage) encompasses the states of anisogamy (compare Figure 5b with Figure 4b). Therefore, while these anisogamous states are resistant to the invasion of gametes of different sizes, they are susceptible to the invasion of asexuals (this recapitulates a common theme in the literature on the evolution of sex: it is difficult to see why anisogamous sex can prevail, if asexual females exist as competitors). Macrogametes are sufficiently large in these states that the contribution by microgametes towards zygote size is negligible. Macrogametes can therefore turn asexual without loss of viability, and once the asexual type invades, selection drives the towards the optimal cell size in isolation (i.e., $m^{*}=\beta_{p}$, see dashed lines in Figure 5).

In contrast to the anisogamous states described above, the isogamous state is, perhaps surprisingly, resistant to the invasion of asexuals. This leads to a rather remarkable situation: despite the fact that we are still in a parameter regime where we assume zygotes have a harder time surviving than parthenogens ($\beta_{z}>\beta_{p}$), e.g., because some of them have fused with genetically incompatible gametes or any other costs of sex, the mere fact that isogamy permits zygotes to be double the size of lone parthenogens can stabilize sexual reproduction. This, of course, assumes that attempts to turn asexual are made difficult by the newly arisen asexual lineage producing suboptimally small cells for parthenogenetic development. This difficulty reflects our assumption of one mutation arising at a time, i.e., a mutant cannot simultaneously change both its mode of reproduction and the size of its progeny. We consider this a potential real feature in many biological systems rather than an artificial constraint imposed by our modelling choices; transitions to asexuality are indeed often difficult when the rest of the life cycle has a past of being selected to perform well under sexual rather than asexual reproduction [49,50,51].

## 4. Discussion

Our model complements a very valuable recent contribution to the literature [25] and shows its results to be structurally robust, in that we recover very similar findings regarding the evolution of anisogamy and isogamy, despite modelling the difference between parthenogenetically and sexually produced progeny in a different way. We explore these similarities and differences below.

### 4.1. Relation to “Evolution of Anisogamy in Organisms with Parthenogenetic Gametes”

As addressed, the model considered in [25] is similar to ours though implemented in a slightly different way; while a Vance survival function is still used for the survival of both zygotes and parthenosporophytes, the same exponent is used in this function for both cases (i.e., $\beta_{z}=\beta_{p}$ in our notation, see Equation (Equation 2) and (Equation 3)). Differences in the survival probability of parthenosporophytes are instead encoded by a multiplication of this factor by a constant 1≥c≥0. Note that this enforces that the survival probabilities of parthenosporophytes are strictly less than those of zygotes. Conversely our approach allows us to consider the distinct regime of parthenosporophyte advantage.

One regime of our model that we do not explore, but that is addressed in [25], is the regime of gamete limitation (i.e., short fertilisation period). Interestingly in this regime, [25] uncovers dynamics that are qualitatively similar to those that we identify under gamete competition (i.e., infinite fertilisation period) but with parthenogenic advantage; the coexistance of stable anisogamous and stable isogamous states (compare our Figure 4a with Figure 3F in [25]). However, the reason for this similar outcome in disparate regimes seems intuitively clear; with gametes of both mating types passing through the parthenogenic route under gamete limitation, there is an increased selective pressure to be well-adapted to the parthenogenic route in [25] that is similar to the selective pressure emerging from parthenogenic advantage in our model. An interesting direction for further work would be to combine these approaches and look at the effect of parthenogenic advantage in the regime of gamete limitation.

Although [25] did not analytically investigate the effect of biased sex ratios analytically, their simulations suggested this had little qualitative effect on their results. Our mathematical analysis indicates a reason for this consistency; under the parthenogenic disadvantage implicitly assumed in [25], differences in the sex ratio can be relatively minor, and merely lead to slight deviations in evolutionary trajectories, as we show in Figure 4a. However, allowing for parthenogenic advantage, as in our model, leads to the emergence of dynamics not covered in [25], namely two routes to asexuality. These routes differ in how big of a threat they are to the maintenance of sex, in a manner that is dependent on whether sex is performed with isogamous or anisogamous gametes.

### 4.2. Two Routes to Asexuality

One of the routes to asexuality involves loss of one of the mating types, leaving the parthenogenetic route the only way to reproduce for the remaining type. This phenomenon relies on overabundance of parthenogenetically developing gametes being able to displace sexuals due to there only being a finite number of ’slots’ available for development into adults in a density-regulated population (i.e., the adult population is of finite size). This approach makes our work link with models of stochastic loss of mating types in finite populations [22,23,24,52,53,54], which previously have been built for isogamous systems only. Here, we have shown that under anisogamy there is potential for the mating type producing smaller gametes to outcompete a type producing larger gametes. This prediction is based on budget constraints which predicts an overabundance of microgametes relative to macrogametes, and if both are able to grow parthenogenetically into adults, the macrogametes may become extinct.

The above prediction that microgametes ’win’ requires that the abundance asymmetry, which favours microgametes, overrides the survival asymmetry, which favours macrogametes, should either attempt parthenogenesis. While we kept the biologically plausible assumption of better macrogamete performance in our model (survival always increases with size, though with potential differences between parthenogenetic and sexually produced progeny), we only considered cases with particularly strong effects of the abundance asymmetry, due to our assumption that all possible sexual fusions have had time to occur before parthenogenetic development begins (the regime of gamete competition). Shorter mating time windows (i.e., moving towards gamete limitation) would allow macrogametes express their superior survival, and the surviving parthenogenetic population might under these conditions result from the macrogamete-producing mating type. Allowing for such dynamics could prove an interesting route for further investigation, and perhaps help explain empirically observed sex ratios in the brown algae, which are typically female biased [32].

However, should one observe a sexual population containing just a single “sex” (i.e., ancestral mating type chromosome) in real life that is a candidate for the above process having happened, we predict difficulties in establishing whether it came about via the micro- or macrogamete route (in other words, did the displaced and now extinct mating type have larger or smaller gametes than the prevailing one). In either case, the remaining mating type is predicted to evolve towards an identical size for its progeny, namely that which is optimized for asexual life cycles. Further, should microgametes be the relevant route, it is clear that at no stage in their past evolutionary trajectory can they have been too small to develop on their own, or have stopped carrying any of the essential components of cytoplasm for further development, in the expectation that the macrogamete will provide them (since any such stage would have blocked the parthenogenetic route for them). For this reason, both micro- and macrogametes will give rise to rather similar asexual lineages, if the loss of mating types is the causal route.

We also derive explicit contrasts between micro- and macrogamete routes to asexuality for the case where asexuality is not a ‘backup option’ that is employed when a sexual fusion failed to happen, but is instead caused by a mutation that makes an individual and its progeny wholly parthenogenetic. Macrogametes survive this type of transition better than microgametes do, which might create an a priori expectation that macrogametes are better able to capitalize on such a mutation. However, this expectation is not always borne out by explicit modelling. Since the quality–quantity tradeoff predicts success to be a function of both components, and parthenogens of microgamete origin can compensate their meagre quality by being produced in large quantities, they may succeed too in making a mutant parthenogen overtake the entire population.

In our example of Figure 5a, a case where parthenogens suffer an intrinsic disadvantage compared with sexual zygotes (the case that [25] consider the more likely one), macrogametes are as a whole unable to invade while microgametes may still do so; however, the regions from where such an invasion occur are themselves not equilibria for gamete sizes, and a system will only pass through these particular gamete sizes in a transitory manner. In the case where parthenogenesis offers survival advantages, there are parameter regions that permit asexual invasion via macrogametes (including cases with stable anisogamy, where stability refers to gamete sizes but not to resistance against asexual invasion), other regions that permit the same to happen via microgametes, a region from which either can invade, and also a region—that includes stable isogamy—where neither type of asexuality can invade. The last fact is intriguing, as isogamous sex is stable against two kinds of assault: alternative gamete sizes and also one mating type turning asexual. This is in spite of the fact that asexuals, were they to produce gametes as large as zygotes, have a survival advantage over sexuals in this scenario.

### 4.3. Relation to Empirical Systems

Finally, it is important to review the behaviour of our model in the context of known empirical results. Parthenogenesis has been observed in isogamous species, as well as in the females of mildly anisogamous species, and, less commonly, also in the males of mildly anisogamous species (see Table 2 in [25]). Our model allows for all these possibilities, and aims to predict what their evolutionary consequences are. While our model predicts the existence of a stable isogamous state, it does not predict mild anisogamy to be stable. Such a state might emerge in other regimes to those we have explored here (e.g., under gamete limitation [25]) or with additional modelling considerations (e.g., gamete mortality in the fertilisation pool [7]). Both these extensions provide interesting avenues for future investigation.

Perhaps the most obvious situation that appears rare in the empirical literature (see Table 1 in [32] for examples) is strong anisogamy combined with male parthenogenesis. Our model explains why this outcome is unlikely: the survival probability of tiny microgametes is exceedingly small should they attempt developing without syngamy (e.g., $10^{-29}$ and $10^{-16}$ for the stable anisogamous states depicted in Figure 2c and d, respectively). A detailed look at the results of [34] makes it clear that some form of selection other than on gamete size must be acting on the propensity for parthenogenic reproduction, as exemplified by a strain of giant kelp that is genetically male, but able to develop parthenogenically irrespective of their size (10 μm-40 μm), with gamete sizes encompassing the entire spectrum of wild type males (5 μm-10 μm) as well as females (35 μm-48 μm). We modelled parthenogenesis as occurring whenever syngamy did not occur; a detailed look at the strain-dependent propensity for parthenogenesis presents a very interesting extension to our model, but would clearly require the inclusion of opportunity costs or other tradeoffs. Given that such tradeoffs have been reported for *Ectocarpus* [31], this is a promising route for future theoretical investigations.

## 5. Conclusions

As a whole, our work joins a recent recognition [25] that trajectories of anisogamy can follow different trajectories from the classic ones when informed by certain real life features of sexual reproduction: sex is often facultative, and this may cause interesting consequences via the possibility that one mating type proliferates asexually. Evolution of sex itself, especially the maintenance of sexual fusions when asexuality is an option, can respond to mating type ratios and gamete size considerations in manners that are not reducible to simple statements such as anisogamy predicting a twofold cost of sex (a number that is in any case only valid given certain simplifying assumptions [41,42]). Instead, or in addition, the predicted course of evolution can be greatly impacted by the ecology of the quality–quantity tradeoff [9,10,11]. Although the dynamics we uncover are clearly more complicated than any simple heuristics that relate the likelihood of a certain mating type winning to how close it is to optimal traits with respect to the quality–quantity tradeoff when reproduction occurs parthenogenetically, such considerations clearly play a part in understanding what transitions are possible.

## Figures and Tables

**Figure 1 cells-10-02467-f001:**
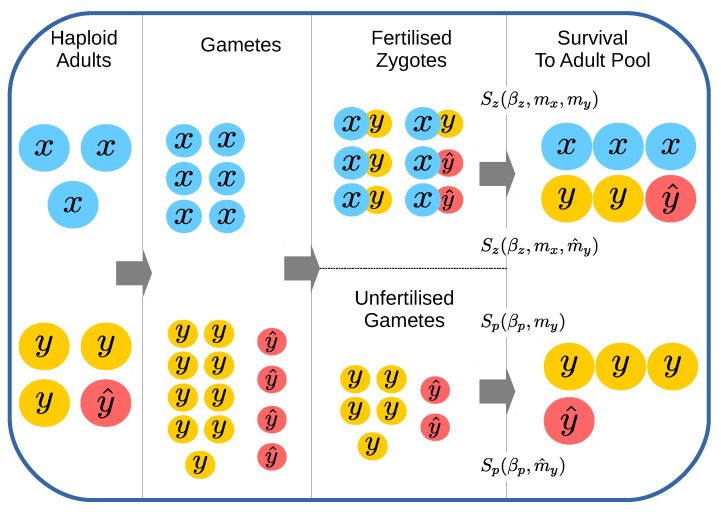
Schematic of Model. Haploid adults divide to form gametes at the start of a generation (see Section 2.1.1). Gametes of opposite mating types (*x* and *y*) fuse to form zygotes, while a number of gametes remain unfertilized in the zygote pool (see Section 2.1.2). Unfertilized gametes form parthenosporophytes, capable of parthenogenic development. Zygotes and parthenosporophytes survive according to to independent survival functions (see Section 2.1.3) to produce a number of haploid adults in the subsequent generation (see Section 2.1.4). Mutant mating types (here $\hat{y}$) produce gametes of a differing size to their ancestors, but inherit their self-incompatibility properties (see Section 2.2.1).

**Figure 2 cells-10-02467-f002:**
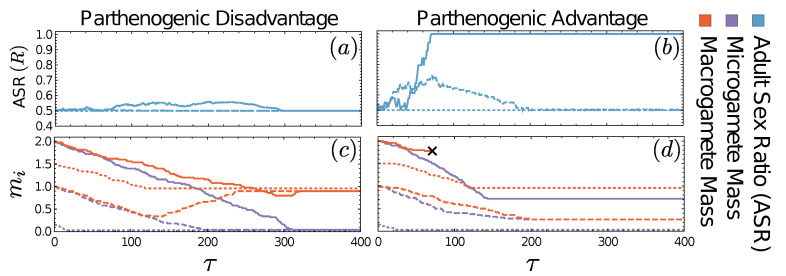
Two examples of evolutionary dynamics without mutations to asexuality. Panels (**a**,**c**): Parthenogenetic disadvantage with $\beta_{p}=1.3>\beta_{z}=1$. Panels (**b**,**d**): Parthenogenetic advantage with $\beta_{p}=0.7<\beta_{z}=1$. The different line types correspond to different initial conditions: solid line, $\left(m_{x},m_{y}\right)=\left(2,2\right)$; dashed lines, $\left(m_{x},m_{y}\right)=\left(1,1\right)$; dotted lines, $\left(m_{x},m_{y}\right)=\left(1.5,0.2\right)$. (**a**,**b**) The adult sex ratio (ASR), i.e., the proportion of adults producing microgametes, deviates little from an even (50:50) ratio in (**a**) but has alternative stable equilibria in (**b**), including 100% microgamete production and extinction of the macrogametes. In (**a**,**c**), all starting conditions lead to the same evolutionary stable anisogamous state with minimally small microgametes $m_{y}=\left|\delta m\right|$ and $m_{x}=1$. In (**b**,**d**), different initial conditions lead to distinct evolutionary stable states: $\left(m_{x},m_{y}\right)=\left(2,2\right)\to \left(\times ,0.7\right)$; $\left(m_{x},m_{y}\right)=\left(1,1\right)\to \left(0.4,0.4\right)$; $\left(m_{x},m_{y}\right)=\left(1.5,0.02\right)\to \left(1,|\delta m|\right)$, where crosses denote extinction of the macrogametic type. Other parameters are ϕ=100 and |δm|=0.02 in all panels.

**Figure 3 cells-10-02467-f003:**
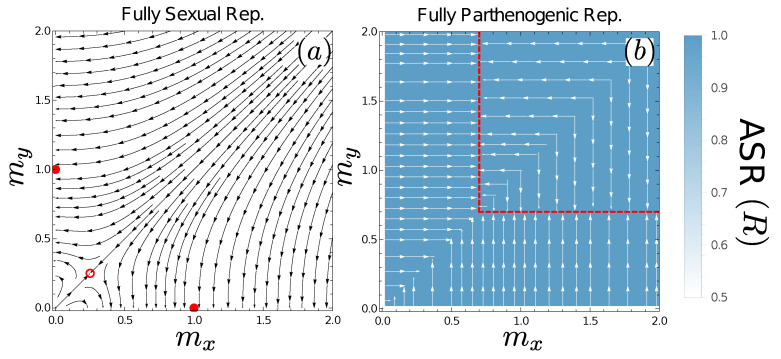
Evolutionary Dynamics under Fully Sexual or Fully Parthenogenic Reproduction. Arrows show the evolutionary trajectories in the limit |δm|→0 (see Equations (Equation 5) and (Equation 6) for panel (**a**) and Equation (Equation 7)) for panel (**b**)). Open red circles give unstable equilibrium points of the evolutionary dynamics, red disks stable equilibrium points, and dashed red lines stable manifold points. The colour (heatmap) indicates the ASR; the heatmaps do not show any responses to gamete sizes because ASR is always 0.5 under fully sexual reproduction and 1 under fully parthenogenetic reproduction. The heatmaps are uniform (and thus relatively uninteresting) but are given to enable direct comparison to later figures.

**Figure 4 cells-10-02467-f004:**
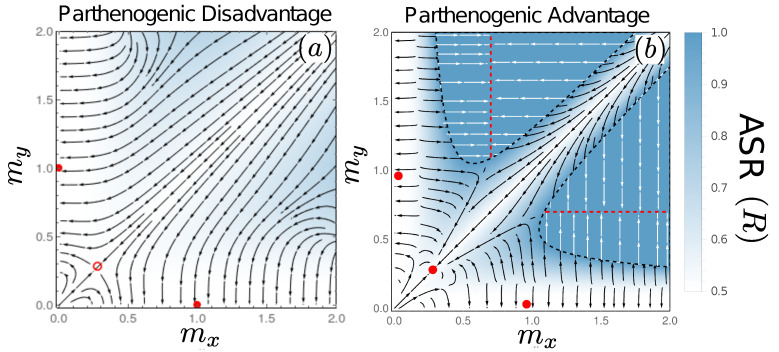
Evolutionary Dynamics under Mixed Sexual and Parthenogenic Reproduction. Evolutionary dynamics when both mating types are present are plotted as gray arrows (see Equations (Equation 9) and (Equation 10)) while evolutionary dynamics when only one mating type is present are plotted as white arrows (Equation (Equation 7). When $\beta_{p}>\beta_{z}$ (panel (**a**)) isogamy is unstable (open red circle) and anisogamy is stable (solid red disks). Certain regions of state space are biased towards higher adult sex ratios of the microgamete (blue overlay). When $\beta_{z}>\beta_{p}$ (panel (**b**)) isogamy and anisogamy are both stable states (solid red disks) and regions emerge where macrogamete producers are driven to extinction (deeper blue region) and for which microgamete producers evolve to producing “gametes” of mass $\beta_{p}$ (dashed red lines). Parameters used are $\beta_{z}=1$, $\beta_{p}=1.3$ (panel (**a**)) and $\beta_{z}=1$, $\beta_{p}=0.7$ (panel (**b**)). These panels are plotted over a larger region of state space in Figure A2.

**Figure 5 cells-10-02467-f005:**
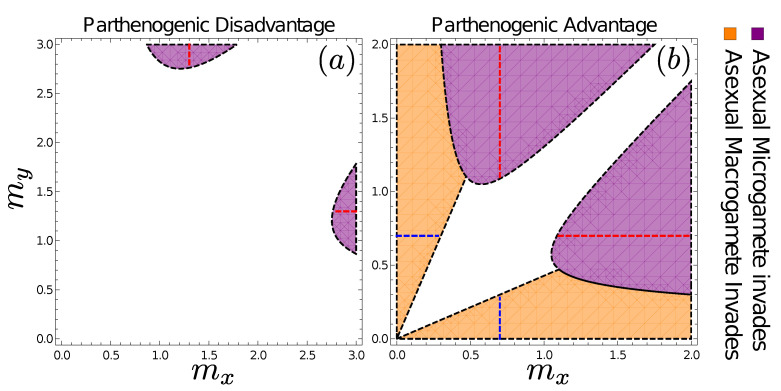
Invasion potential of asexuals when (**a**) $\beta_{p}=1.3>\beta_{z}=1$ and (**b**) $\beta_{p}=0.7<\beta_{z}=1$. Asexuality invades more easily, and can do so via mutations in either type, in (**b**) where parthenogenesis brings about a survival advantage. Microgametes can do so in (**b**), in the purple regions, which are coincident with the regions in which microgametes can drive macrogametes extinct in the absence of mutations to asexuality (see Figure 4). Note and that the plane contains regions where *y* is the macrogamete producer (above the diagonal) as well as *x* being the macrogamete producer (below the diagonal); also note the sightly larger range of the $m_{x}-m_{y}$ plane depicted in (**a**), to make the microgamete invasion region visible. Isogamous sex occurs along the diagonal, where isogamous sex, should it be stable against deviating gamete sizes (as it is in (**b**)), is also protected against invasion of asexual mutants as indicated by the white region in (**b**). Following the successful invasion of an asexual, the resulting evolutionary dynamics are governed by Equation (Equation 7), taking the population to a state in which parthenogens produce gametes of size $m_{a}=\beta_{p}$, denoted by red-dashed lines for asexuals derived from microgametes and blue-dashed lines for asexuals derived from macrogametes.

## Data Availability

Analysis, simulation code and data arising from simulations can be found in the repository https://github.com/gwaconstable/parthEvoAnisogamy, created 25 May 2021.

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
