# Peer review of "Parthenogenesis and the Evolution of Anisogamy"

_cells, 2021, doi:10.3390/cells10092467_

Round 1

Reviewer 1 Report

Interesting debate on the models of the potential evolution of parthenogenesis. As authors state as first, this is a complementary-further reasoning about the evolution of anisogamy in the possibility of parthenogenetic reproduction, despite or taking into into consideration the fact  that sex is facultative in many relevant taxa (e.g. algae) that harbour both anisogamous and isogamous species.

I appreciated their approach. The only thing that I would like to read at the end of the manuscript, in the Discussion section, is a deeper comment by comparing their results/conclusions with those presented in [1]

Author Response

Does the introduction provide sufficient background and include all relevant references? Yes

Is the research design appropriate? Yes

Are the methods adequately described? Yes

Are the results clearly presented? Yes

Are the conclusions supported by the results? Can be improved

Comments and Suggestions for Authors:

Interesting debate on the models of the potential evolution of parthenogenesis. As authors state as first, this is a complementary-further reasoning about the evolution of anisogamy in the possibility of parthenogenetic reproduction, despite or taking into into consideration the fact that sex is facultative in many relevant taxa (e.g. algae) that harbour both anisogamous and isogamous species.

I appreciated their approach. The only thing that I would like to read at the end of the manuscript, in the Discussion section, is a deeper comment by comparing their results/conclusions with those presented in [1]

Response and Changes from Authors:

 We thank the reviewer for their positive assessment of our work. We also agree that there is indeed room in the discussion for a more detailed comparison of our results with those discussed in [1]. In response we have added three paragraphs to the Discussion section, detailing differences in the model framework (L549-L556), different regimes covered by the respective papers that nevertheless lead to the same qualitative outcome (L557-L568), and parameter regimes covered by both papers but with complementary analysis (L569-L577).

Reviewer 2 Report

This manuscript addresses an interesting area of research and is well written and well organised.

I have two fundamental issues with the work:

  1. The authors should make it clear that they are assuming that gamete limitation plays a role in the evolution of anisogamy and related phenomena. This approach has been promoted by Lehtonen and Kokko (2011) but is of limited value since gamete limitation is expected to be rare in the organisms of interest, and when it occurs, gamete competition is a much more important force even when gamete limitation is severe (Parker and Lehtonen 2014). Furthermore, the focus on gamete limitation derives from group selection theory, which should be justified explicitly.
  2. The organisms that this work is modelling are cyclically parthenogenetic, but the model does not represent this class of lifecycle.

Therefore, I believe that for this work to be relevant to explaining any natural phenomena, the two above issues should be addressed. Furthermore, to determine the relevance of the model it would be necessary to at least discuss the model’s predictions in relation to natural systems.

Lehtonen, J., and H. Kokko. 2011. Two roads to two sexes: Unifying gamete competition and gamete limitation in a single model of anisogamy evolution. Behavioral Ecology and Sociobiology 65:445-459.

Parker, G. A., and J. Lehtonen. 2014. Gamete evolution and sperm numbers: sperm competition versus sperm limitation. Proceedings of the Royal Society B: Biological Sciences 281.

Author Response

Does the introduction provide sufficient background and include all relevant references? Can be improved

Is the research design appropriate? Can be improved

Are the methods adequately described? Yes

Are the results clearly presented? Yes

Are the conclusions supported by the results? Can be improved

Comments and Suggestions for Authors:

This manuscript addresses an interesting area of research and is well written and well organised. I have two fundamental issues with the work:

  1. The authors should make it clear that they are assuming that gamete limitation plays a role in the evolution of anisogamy and related phenomena. This approach has been promoted by Lehtonen and Kokko (2011) but is of limited value since gamete limitation is expected to be rare in the organisms of interest, and when it occurs, gamete competition is a much more important force even when gamete limitation is severe (Parker and Lehtonen 2014). Furthermore, the focus on gamete limitation derives from group selection theory, which should be justified
  2. The organisms that this work is modelling are cyclically parthenogenetic, but the model does not represent this class of lifecycle.

Therefore, I believe that for this work to be relevant to explaining any natural phenomena, the two above issues should be addressed. Furthermore, to determine the relevance of the model it would be necessary to at least discuss the model’s predictions in relation to natural systems.

Lehtonen, J., and H. Kokko. 2011. Two roads to two sexes: Unifying gamete competition and gamete limitation in a single model of anisogamy evolution. Behavioral Ecology and Sociobiology 65:445-459.

Parker, G. A., and J. Lehtonen. 2014. Gamete evolution and sperm numbers: sperm competition versus sperm limitation. Proceedings of the Royal Society B: Biological Sciences 281.

Response and Changes from Authors:

Here, we can see that we need to write more carefully to avoid misunderstandings. Gamete limitation refers to a setting where even the less numerous gamete type (typically, macrogametes) does not achieve 100% success being fertilized. Here, our main focus is on situations where the fertilisation period is infinitely long (i.e. φ→∞), where all macrogametes will be fertilized and only microgametes

have any risk of failing – indeed, this imposes strong competition among the microgametes, and no failures among the macrogametes, which is precisely the setting of gamete competition, not gamete limitation.

We have now addressed these assumptions more explicitly at the opening to the Results section, as well as recapping on this in Discussion section, in the hope that misunderstandings can be avoided.

“The first of these restrictions means that we restrict our analysis to the regime of gamete competition (under which all macrogametes are fertilized and there is competition for fertilizations among the microgametes) and ignore the regime of gamete limitation (under which fertilization is

inefficient and macrogametes do not achieve 100% fertilization success) [1,8,47].” - L334-L339

“One regime of our model that we do not explore, but that is addressed in [1], is the regime of gamete limitation (i.e. short fertilisation period). … An interesting direction for further work would be to combine these approaches and look at the effect of parthenogenic advantage in the regime of gamete limitation.” L557-L568

We would also like to take issue with the claim that gamete limitation relies on group selection. This is admittedly not a very relevant point to make here, as gamete limitation is only permitted in principle in our model (i.e. we could have focused on small φ values but did not choose to do so). Still, it is important for readers of this literature to know the difference between early group selection models (Kalmus 1932 and Scudo’s work in the 1960s) and the modern way of analyzing gamete limitation, which does not assume that an entire group of organisms maximizes success together, but instead derives the success of a deviating mutant (e.g. an individual female) in a group of others. This is strictly individual selection, which is derived taking into account – correctly – that the success of an individual depends on how many competitors in a local group it has to deal with.

While we agree that this is an important distinction, we note that this historical perspective has already been explored in earlier papers that we cite, e.g. [8], who write:

“Finally, it is also worth commenting on the oldest model of anisogamy, Kalmus (1932). This model was published decades before biologists were trained to realize that individuals or their genes can be in conflict with each other. Kalmus' model has been largely dismissed as group selectionist, yet this model is surprisingly relevant to this day in the special case of A=1. When there is only one individual of each mating type present, both “paternal” and “maternal” fitness are maximized when parents behave in a way that maximizes the number of viable zygotes, even though each parent is acting “selfishly” in its own best interests.”

We therefore believe that restating these facts here would distract from the core arguments of the paper, as well a potentially confusing those new to the field.

Regarding the comment about cyclical parthenogenesis, we believe that the impression of the reviewer that our model fails to capture reality is a little too pessimistic. The way we interpret the reviewer’s comment, the issue is whether our assumption, that asexual reproduction can occur when a gamete has failed to meet a partner, is a realistic one. We believe it is. As described in reference [31] https://www.ncbi.nlm.nih.gov/pmc/articles/PMC6592573/:

“The haploid-diploid life cycle of the model brown alga Ectocarpus involves an alternation between a haploid gametophyte and a diploid sporophyte. Superimposed on this sexual cycle, an asexual,

parthenogenetic cycle has been described for some Ectocarpus strains (Fig 1A) [19,22]. In this parthenogenetic cycle, gametes that fail to meet a partner of the opposite sex develop into haploid parthenosporophytes. These parthenosporophytes are indistinguishable morphologically from diploid sporophytes [19]. Parthenosporophytes can produce gametophyte progeny to return to the sexual cycle through two mechanisms: 1) endoreduplication during development to produce diploid cells that can undergo meiosis or 2) individuals that remain haploid can initiate apomeiosis [19,22].” (emphasis added)

Finally, the reviewer asks that we discuss the model’s predictions in relation to natural systems. We note that similar issues were also raised by Reviewers 3 and 4, and so we have sought to redress this balance in the discussion by adding two new paragraphs to the discussion:

“Finally, it is important to review the behaviour of our model in the context of known empirical results. Parthenogenesis has been observed in isogamous species, as well as in the females of mildly anisogamous species, and, less commonly, also in the males of mildly anisogamous species (see Table~2 in [1]). Our model allows for all these possibilities, and aims to predict what their evolutionary consequences are. While our model predicts the existence of a stable isogamous state, it does not predict mild anisogamy to be stable. Such a state might emerge in other regimes to those we have explored here (e.g. under gamete limitation [1]) or with additional modelling considerations (e.g. gamete mortality in the fertilisation pool [8]). Both these extensions provide interesting avenues for future investigation.

Perhaps the most obvious situation that appears rare in the empirical literature (see [32], Table~1, for examples) is strong anisogamy combined with male parthenogenesis. Our model explains why this outcome is unlikely: the survival probability of tiny microgametes is exceedingly small should they attempt developing without syngamy (e.g. $10^{-29}$ and

$10^{-16}$ for the stable anisogamous states depicted in Figure~\ref{fig_illustrative}c~and~d respectively). A detailed look at the results of [34] makes is clear that some form of selection other than gamete size must be acting on the propensity for parthenogenic reproduction, as exemplified by a strain of giant kelp that is genetically male, but able to develop parthenogenically irrespective of their size (10{\textmu}m-40{\textmu}m), with gamete sizes encompassing the entire spectrum of wild type males (5{\textmu}m-10{\textmu}m) as well as females (35{\textmu}m-48{\textmu}m). We modelled parthenogenesis as occurring whenever syngamy did not occur; a detailed look at the strain-dependent propensity for parthenogenesis presents a very interesting extension to our model, but would clearly require the inclusion of opportunity costs or other trade-offs. Given that such trade-offs have been reported for

\textit{Ectocarpus}[31], this is a promising route for future theoretical investigations.” - L642- L666

Reviewer 3 Report

This manuscript investigates the interaction of parthenogenesis and the evolution of anisogamy. They explore the situation in which microgametes proliferate parthenogenetically, which leads to a proliferation of parthogenetic reproduction, creating a positive feedback loop and leading to asexuality. Additionally, they find a parameter space in which isogamous populations resist invasion by asexual individuals.

I find a major issue with the model that allows for microgametes (sperm) to reproduce parthenogenetically- I am not aware of any such organisms existing. As such, the authors should provide examples of the biological relevance of this model. The model is inspired by Ectocarpus, but as they detail on page 3, Ectocarpus is primarily isogamous, not anisogamous. The authors should go into more detail justifying the ecological relevance of their model, as well as the applicability in the Discussion section.

Author Response

Does the introduction provide sufficient background and include all relevant references? Yes

Is the research design appropriate? Yes

Are the methods adequately described? Yes

Are the results clearly presented? Yes

Are the conclusions supported by the results? Yes

Comments and Suggestions for Authors:

This manuscript investigates the interaction of parthenogenesis and the evolution of anisogamy. They explore the situation in which microgametes proliferate parthenogenetically, which leads to a proliferation of parthogenetic reproduction, creating a positive feedback loop and leading to asexuality. Additionally, they find a parameter space in which isogamous populations resist invasion by asexual individuals.

I find a major issue with the model that allows for microgametes (sperm) to reproduce parthenogenetically- I am not aware of any such organisms existing. As such, the authors should provide examples of the biological relevance of this model. The model is inspired by Ectocarpus, but as they detail on page 3, Ectocarpus is primarily isogamous, not anisogamous. The authors should go into more detail justifying the ecological relevance of their model, as well as the applicability in the Discussion section.

Response and Changes from Authors:

Here, we would like to point out that ‘sperm’ (using the word in the sense of ‘very small microgametes’) are indeed unlikely to achieve parthenogenesis. The reason that we wrote

“While it is tempting to emphasize the wonderful weirdness of this result by stating that ‘sperm drove eggs extinct’, this mental image misrepresents the dynamics. The microgametes obviously were at no point along the evolutionary trajectories so small that their viability for parthenogenetic reproduction was too severely compromised, […]”

was precisely to avoid a reader interpreting the model as a claim that actual sperm (microgametes in systems where anisogamy is strong) will be able to take this route. The dynamics that emerges from our model is indeed a prediction that they will not do so (simply because survival is too unlikely in those cases). Instead, we show that microgametes under mild anisogamy ratios – i.e. in the inner parts of a figure such as fig. 4, not near the axis where one gamete is really small – could maintain the ability to do so, and there is indeed evidence that this is possible. See

https://nph.onlinelibrary.wiley.com/doi/10.1111/nph.17582 (and https://phys.org/news/2021-07- genetically-male-strain-giant-kelp.html for a pop-sci version):

the giant kelp demonstrates unequivocally that the U-chromosome is not required to initiate the female developmental program

and

The genetically male kelp is almost indistinguishable from a female, and most strikingly, can even go through parthenogenesis, a process that usually is exclusive to females

This, to us, suggests that it is an entirely valid exercise to ask how small microgametes can get before the route to asexuality becomes completely blocked. We now refer to this in the introduction:

“Interestingly, a recent study in giant kelp (which are closely related to

\textit{Ectocarpus}~\cite{phillips2008}), identified a genetically male strain that was capable of parthenogenesis ~\cite{muller2021} via gametes that were on average half the size of those produced by females. Taken together, this suggests that at least in principle, the propensity of either sex to reproduce parthenogenically may be more labile than commonly assumed.” -

L107-114.

However, as the reviewer correctly points out, this does not seem to be a valid route for tiny sperm. We thoroughly agree with this, and our model does too; but we do note there is an important region where gamete sizes are much more similar where either sex can, in principle, perform parthenogenesis. We now write:

“Perhaps the most obvious situation that appears rare in the empirical literature (see [32], Table~1, for examples) is strong anisogamy combined with male parthenogenesis. Our model explains why this outcome is unlikely: the survival probability of tiny microgametes is exceedingly small should they attempt developing without syngamy (e.g. $10^{-29}$ and

$10^{-16}$ for the stable isogamous states depicted in Figure~\ref{fig_illustrative}c~and~d respectively). ” - L663-658

Finally, when the reviewer writes that Ectocarpus is ‘primarily isogamous’, one could take this as a recommendation to only focus on deriving the consequences of isogamy in the model. In our interpretation, however, it is not really a flaw of our model that we are able to describe the dynamical consequences under isogamy, mild anisogamy, as well as strong anisogamy. Note that we do also evaluate what is required for a system to stay in an isogamous region (lines 462-468, 528-536 and 637- 638 are devoted to this issue).

Reviewer 4 Report

Review of the manuscript

Parthenogenesis and the Evolution of Anisogamy

The manuscript cells-1254227, dealing with the that classic models for the relationship between progeny size which may differ between parthenogenetically and  sexually produced progeny,  is well written paper with properly performed analyses. I only wondering, why the brown algae Ectocarpus has been a model for these studies.. Maybe author should compare this results with results on different kind of species.

Overall, the text, methodology and figures are correct. Due to the high importance of the research conducted with all used tools, the manuscript should be accepted for publication even in current form. However, in my opinion, from the embryological and genetic point of view , described process is full of unexpected, random events that we cannot enter into the model… But it’s a very interesting work and a way of considering developmental processes, which should be accessible to more readers.

Author Response

Does the introduction provide sufficient background and include all relevant references? Yes

Is the research design appropriate? Yes

Are the methods adequately described? Yes

Are the results clearly presented? Yes

Are the conclusions supported by the results? Can be improved

Comments and Suggestions for Authors:

The manuscript cells-1254227, dealing with the that classic models for the relationship between progeny size which may differ between parthenogenetically and sexually produced progeny, is well written paper with properly performed analyses. I only wondering, why the brown algae Ectocarpus has been a model for these studies.. Maybe author should compare this results with results on different kind of species.

Overall, the text, methodology and figures are correct. Due to the high importance of the research conducted with all used tools, the manuscript should be accepted for publication even in current form. However, in my opinion, from the embryological and genetic point of view , described process is full of unexpected, random events that we cannot enter into the model… But it’s a very interesting work and a way of considering developmental processes, which should be accessible to more readers.

Response and Changes from Authors:

Thank you for this very kind evaluation. Our choice of Ectocarpus should be seen as a source of inspiration rather than the model restricting its view to one genus only. By virtue of being particularly well studied, Ectocarpus offers direct evidence for many of the processes we envisage (see e.g. our reply to reviewer 3 which shows the importance of being able to show direct evidence for certain modelling assumptions). We kept the empirical section rather brief in our 1st version, because ref. [1] did such a thorough job in reviewing the evidence. We have now added some more detail from the empirical literature, as well as Coelho’s recent paper (see final response to Reviewer 2 and response to Reviewer 3):

“Interestingly, a recent study in giant kelp (which are closely related to

\textit{Ectocarpus}~\cite{phillips2008}), identified a genetically male strain that was capable of parthenogenesis ~\cite{muller2021} via gametes that were on average half the size of those produced by females. Taken together, this suggests that at least in principle, the propensity of either sex to reproduce parthenogenically may be more labile than commonly assumed.” -

L107-114.

“Finally, it is important to review the behaviour of our model in the context of known empirical results. Parthenogenesis has been observed in isogamous species, as well as in the females of mildly anisogamous species, and, less commonly, also in the males of mildly anisogamous species (see Table~2 in [1]). Our model allows for all these possibilities, and aims to predict what their evolutionary consequences are. While our model predicts the existence of a stable isogamous state, it does not predict mild anisogamy to be stable. Such a state might emerge in other regimes to those we have explored here (e.g. under gamete limitation [1]) or with additional modelling considerations (e.g. gamete mortality in the fertilisation pool [8]). Both these extensions provide interesting avenues for future investigation.

Perhaps the most obvious situation that appears rare in the empirical literature (see [32], Table~1, for examples) is strong anisogamy combined with male parthenogenesis. Our model explains why this outcome is unlikely: the survival probability of tiny microgametes is exceedingly small should they attempt developing without syngamy (e.g. $10^{-29}$ and $10^{-16}$ for the stable anisogamous states depicted in Figure~\ref{fig_illustrative}c~and~d respectively). A detailed look at the results of

[34] makes is clear that some form of selection other than gamete size must be acting on the propensity for parthenogenic reproduction, as exemplified by a strain of giant kelp that is genetically male, but able to develop parthenogenically irrespective of their size (10{\textmu}m-40{\textmu}m), with gamete sizes encompassing the entire spectrum of wild type males (5{\textmu}m-10{\textmu}m) as well as females (35{\textmu}m-48{\textmu}m). We modelled parthenogenesis as occurring whenever syngamy did not occur; a detailed look at the strain-dependent propensity for parthenogenesis presents a very interesting extension to our model, but would clearly require the inclusion of opportunity costs or other trade-offs. Given that such trade-offs have been reported for \textit{Ectocarpus}[31], this is a promising route for future theoretical investigations.” - L641-665

Round 2

Reviewer 2 Report

The last sentence of the Abstract makes no sense. There must be a word or some punctuation missing. 

In response to my comment that the model depends on gamete limitation, which is hard to justify, the authors admit that it does, but that it is irrelevant since gamete limitation is not analysed. Then they contradict themselves by explaining that gamete parthenogenesis occurs when gametes do not fuse. This makes no sense.

In response to my comment that the lifecycle modelled does not appear to reflect any real lifecycle, the authors point to a single species of brown algae. This does not seem to justify the modelling effort.

Finally, in response to my comment that no testable predictions are made from the model, the authors argue that the model predicts that male gamete parthenogenesis will not evolve in strongly anisogamous species because the microgametes are so small that their probability of survival will be extremely low. This is intuitively obvious and could be deduced from simple biological considerations without the enormous modelling effort that has gone into this work.

I think the authors have developed a rather complicated model that includes many possible processes, but with little consideration for their biological plausibility. The result is an enormous amount of work that does not extend our understanding of the evolution of anisogamy. 

Reviewer 3 Report

The reviewers have addressed my previous comments and the paper should be accepted.

Author Response

We're please that the reviewer feels their comments have been addressed, and thank them for the time spent reading and giving us feedback on our manuscript.